# Hybrid evolutionary machine learning model for advanced intrusion detection architecture for cyber threat identification

Ankita Sharma[1], Shalli Rani [1]*, Maha Driss[2,3]

1 Chitkara University Institute of Engineering and Technology, Chitkara University, Chandigarh, Punjab, India, 2 Robotics and Internet-of-Things Laboratory, Prince Sultan University, Riyadh, Saudi Arabia, 3 RIADI Laboratory, National School of Computer Sciences, University of Manouba, Manouba, Tunisia

These authors contributed equally to this work.
* Shalli.rani@chitkara.edu.in

**Data Availability Statement:** All data are in the manuscript.

**Funding:** The authors would like to acknowledge the support of Prince Sultan University for paying

## Abstract

In response to the rapidly evolving threat landscape in network security, this paper proposes an Evolutionary Machine Learning Algorithm designed for robust intrusion detection. We specifically address challenges such as adaptability to new threats and scalability across diverse network environments. Our approach is validated using two distinct datasets: BoT-IoT, reflecting a range of IoT-specific attacks, and UNSW-NB15, offering a broader context of network intrusion scenarios using GA based hybrid DT-SVM. This selection facilitates a comprehensive evaluation of the algorithm's effectiveness across varying attack vectors. Performance metrics including accuracy, recall, and false positive rates are meticulously chosen to demonstrate the algorithm's capability to accurately identify and adapt to both known and novel threats, thereby substantiating the algorithm's potential as a scalable and adaptable security solution. This study aims to advance the development of intrusion detection systems that are not only reactive but also preemptively adaptive to emerging cyber threats." During the feature selection step, a GA is used to discover and preserve the most relevant characteristics from the dataset by using evolutionary principles. Through the use of this technology based on genetic algorithms, the subset of features is optimised, enabling the subsequent classification model to focus on the most relevant components of network data. In order to accomplish this, DT-SVM classification and GA-driven feature selection are integrated in an effort to strike a balance between efficiency and accuracy. The system has been purposefully designed to efficiently handle data streams in real-time, ensuring that intrusions are promptly and precisely detected. The empirical results corroborate the study's assertion that the IDS outperforms traditional methodologies.

## 1 Introduction

Significant advancements in the area of information technology have been made possible by the introduction of computational evolution and machine learning. Machine learning models

the Article Processing Charges (APC) of this publication.

**Competing interests:** NO authors have competing interests.

are optimized by evolutionary computing approaches, which draw inspiration from the principles of natural selection [1]. This process increases the models' adaptability and efficiency. As this collaborative effort progresses, a critical concern surfaces: the absolute need for strong cybersecurity protocols to protect the availability, confidentiality, and dependability of these evolutionary computing-based machine learning systems. Since evolutionary computing techniques like genetic programming, GAs, and evolutionary methods were introduced, the subject of machine learning has seen significant change. These algorithms are very good at navigating large solution spaces, adapting to dynamic settings, and improving complex models [2]. Evolutionary computation is necessary to facilitate feature selection and parameter fine-tuning, hence increasing the effectiveness of machine learning applications [3]. Evolutionary computing-based machine learning systems are capable of great things, but they are also prone to errors [4]. Algorithms' cyclical and dynamic nature produces fresh weaknesses that might be taken advantage of by hostile organisations. Notable dangers including adversarial attacks, data poisoning, and model inversion highlight the need of putting in place a thorough cybersecurity strategy [5]. Machine learning models that handle sensitive data and make important decisions, employ evolutionary computation optimisation. The real repercussions of jeopardising these models' validity and anonymity highlight how crucial it is to implement robust cybersecurity defences [6]. Adversarial threats, which include manipulating input data to trick ML systems, become more sophisticated than they were in their initial condition when evolutionary computing is involved [7]. GA rely heavily on selection mechanisms that can, under certain conditions, lead to reduced genetic diversity within the population. This decrease in diversity might cause the algorithm to overfit to anomalies or disturbances in the data, potentially magnifying their impact and leading to less robust solutions. This emphasizes the need of implementing cybersecurity measures that possess the adaptability to identify and counteract constantly evolving threats [8]. Therefore, this study investigates the question: How can evolutionary machine learning algorithms enhance the detection accuracy and adaptability of intrusion detection systems against evolving cyber threats? The cybersecurity landscape is characterized by its constantly evolving and dynamic nature. The simultaneous development and adjustment of cyber threats may make traditional security methods insufficient [9]. Evolutionary computing in the current scenario requires a cybersecurity infrastructure that has the capacity to adapt, identify anomalies, and respond in real-time to emerging threats [10]. The increasing use of machine learning techniques in cybersecurity operations, namely for the purpose of detecting threats, anomalous authentication, and statistical analysis, showcases the complex interconnection between machine learning and cybersecurity [11, 12]. Gathering network datasets involves capturing network activity in the form of packets of data, which might have many attributes [13, 14]. Some characteristics may be unnecessary or insignificant when considering intrusion detection [15]. Moreover, the efficiency of detecting patterns and the computational expenses may be negatively impacted by extensive datasets that include a significant number of characteristics [16]. ML models often use various feature reduction approaches to improve the capacity for learning and reduce computing complexity. The following paper presents a Hybrid Evolutionary Machine Learning Model that is being designed to improve the capabilities of advanced IDS architectures. By leveraging the strengths of EA and ML techniques, this hybrid model aims to provide a more resilient and adaptive solution for cyber threat identification. EA offer robust search capabilities and adaptability, while ML techniques excel in pattern recognition and predictive analytics. The combination of these methodologies results in a powerful tool that can evolve and improve over time, adapting to new and emerging threats in the cybersecurity landscape.

## 1.1 Motivation

The motivation behind this research stems from the increasing complexity of cyber threats and the limitations of existing intrusion detection systems. Traditional IDS often rely on static rule-based approaches, which are insufficient in detecting novel and sophisticated attacks. Furthermore, machine learning-based IDS, while more dynamic, can suffer from issues such as high false positive rates and the inability to adapt to rapidly changing threat environments. Also, conventional models frequently fail to detect intricate or groundbreaking attacks, resulting in high rates of false negatives and the introduction of vulnerabilities in the system. To overcome the limitations of traditional system the proposed approach focuses on enhancement in the detection and management of a wide range of cyber threats, hence reducing the likelihood of security breaches, by highlighting the algorithm's ability to adapt to the threat landscape which motivated us to integrate ML with EC. By combining EA with machine learning, the proposed hybrid model seeks to address these challenges by providing a more adaptive and accurate intrusion detection system. EA can explore a vast search space and optimize detection strategies over time, while machine learning models can continuously learn and update based on new data. This hybrid approach aims to reduce false positives, enhance detection accuracy, and provide a scalable solution.

## 1.2 Our contributions

- This study employs the Bot-IoT and UNSW-NB15 intrusion datasets to evaluate mean fitness using ML classifiers. An innovative fitness function is constructed for the GA to rank features effectively. The feature selection process is guided by GA, with specific parameters tailored for optimal performance.

- To achieve the best possible outcomes, the model integrates the GA with DT and SVM. Additionally, the FPR is incorporated as a parameter in the newly created fitness function to simultaneously reduce FAR and enhance the detection of TPR. A high FPR can lead to many benign activities being incorrectly classified as threats, resulting in unnecessary alerts and operational inefficiencies. This disrupts user trust and increases the operational burden due to the need to investigate each false alarm, thus acting as a penalty on the system's usability and resource management.

- The study develops a genetic algorithm-based feature selection methodology that leverages evolutionary computing for feature selection in intrusion detection systems. The F1-score is used to balance the importance given to both the recall rate and the accuracy rate. The F1 score harmonizes recall and accuracy, addressing potential imbalances that might lead to misleading interpretations of a model's accuracy. The main goal is to enhance the precision and computational capacity of DT and SVM by optimizing parameters and reducing data dimensionality.

There exist distinct security risks and possible breaches in different types of applications such as smart cities, smart industries, smart education etc. Many common attack strategies that specifically target vulnerable domains due to their reliance on linked technology and networks. This includes the following attacks that the network of smart cities and other smart applications are vulnerable to:

- DDoS: This attack target vital infrastructure or public services. The potential vulnerability of municipal communication networks or control systems poses a significant risk to transportation, electrical infrastructures, and emergency services, potentially leading to grave repercussions [17].

- Malware: The use of malicious software to infiltrate municipal systems and get unauthorized access to sensitive data. The collection of large amounts of data by local governments raises concerns about potential privacy violations [18].

- Unauthorized acquisition or breach of personal information: Occurrences of mishandling or improper storage of personal information, which may result in illegal access or identity theft [19]. Efforts to undermine the physical infrastructure include intercepting data transmissions inside the city's communication infrastructure, posing a possible risk to private data. Targeting the infrastructure of smart communities has the capacity to significantly impede operations.

- Attacks on physical devices: Disruption of traffic control systems by means of vandalism, destruction, or harm to surveillance cameras, sensors, or IoT equipment may result in crashes or traffic congestion. The disruption of essential services, such as power or water delivery, due to irregularities or manipulation of the infrastructure [20].

- Eavesdropping: It refers to the deliberate intrusion or surveillance of persons by exploiting or taking control of connected equipment, such as security cameras or smart locks. Engaging in surveillance of smart home communication channels with the purpose of obtaining confidential information or gathering personal insights about the residents. Smart home gadgets that are equipped with IoT technology are vulnerable to malicious exploitation [21].

- Botnet: Exploiting weaknesses in IoT devices to take control of them and possibly provide unauthorized users access to the home network. To initiate botnet attacks that impair internet connection or service availability, exploit compromised devices. The large volumes of data collected by smart home devices raise concerns about potential privacy infringements and spying [22].

- Data tampering: It refers to the deliberate abuse of personally identifiable information collected by intelligent gadgets, such as health monitoring systems and voice assistants. Cybersecurity breach refers to the deliberate act of leveraging vulnerabilities in data-sharing or cloud storage systems with the intention of revealing confidential information to the public. Tampering with neonatal monitors or security cameras undermines privacy and facilitates invasive monitoring [23].

## 1.3 Preliminaries

A feature selection approach is a method used to choose a specific, concise, and precise subset of features from a given set of features. For this study, we have opted for a correlation-based feature selection method, which assesses the significance of the features are evaluated using entropy and information gain [24]. Specifically, ineffective, noisy, and redundant features must be eliminated. An evolutionary method to feature selection is employed, utilizing three separate evolutionary search techniques: Particle Swarm Optimization (PSO) [25], Genetic Algorithm (GA) [26], and Ant Colony Optimization (ACO) [27].

- PSO: This technique involves representing a set of features using particles within a swarm. A number of particles are positioned in a hyperspace, with each particle having a random location $l_i$ and velocity $o_i$. Let $\alpha$ represent the inertia weight constant, while $c_i$ and $s_i$ represent the cognitive and social learning constants, respectively. Let $a_1$ and $a_2$ represent arbitrary integers, pi denote the individual particle i's optimum location, and p indicate the global placement among all particles. Next, the essential guidelines for modifying the position and

velocity of every particle are as shown in (1) and (2):

$$l_i((p+1)) = l_i(p) + o_i((p+1)) \tag{1}$$

$$o_i(p+1) = \alpha o_i(p) + c_1 s_1(p_i l_i(p)) + c_2 s_2(g l_i(p)) \tag{2}$$

- GA: Defining the parameters, such as the initial population, mutation rate, crossover probability, and k value, is crucial in the experiment. This strategy use a chromosome as a representation of a set of characteristics. In a set of features, a binary number is employed to denote the presence (1) or absence (0) of a specific feature. In addition, the subset evaluator employs k-fold cross-validation to assess the input features, and the Goldberg method is commonly employed to obtain the optimal feature set. It is crucial to establish the parameters of the experiment. The parameters encompass the initial population, mutation rate, crossover chance, and k value.

- ACO: This technique utilizes a graph representation, where features are represented as nodes and the identification of the optimal next feature is shown along the periphery. The ultimate feature subset is acquired by an ant search algorithm on the graph. The search halting criterion is the condition that determines when the search process should be terminated and configured to verify a minimal quantity of traversed nodes [27]. Furthermore, to assess the relative importance of different aspects among the currently selected features, a probabilistic transition rule is used for informative purposes. Let k represent the numerical value of ants in total, $P_i^k$ set of non-visited k ants, $m_{ij}$ is the heuristic value of feature j, $O_{ij}$ (t) is pheronome which is virtually considered on edge (i,j) The probability of an ant at feature i being inclined to travel to feature j at time t is as shown in (3):

$$p_{ij}^k = \frac{[O_{ij}(t)]^\alpha . [m_{ij}]^\beta}{\sum_{J_i^k} [O_{ij}(t)]^\alpha . [m_{ij}]^\beta} \tag{3}$$

## 2 Related work

Ensuring the ethical use of machine learning via evolutionary computing requires the incorporation of privacy, accountability, and fairness protocols. By monitoring and preventing unwanted access or data alteration, IDS helps to uphold ethical standards [24]. The selection of features is a basic preprocessing step in ML. Reducing the number of dimensions in the data improves the effectiveness of the categorization process. Researchers have proposed several feature choices for multiple IDS. These strategies are used to classify important characteristics based on various criteria. In the context of IDSs, a group of researchers in Reference [25] developed a hybrid model that combines SVM with GA. This model has the capability to decrease the number of selectable attributes from 41 to 10. The use of GA resulted in the classification of the selected characteristics into three distinct groups based on their respective levels of significance. The attributes that are considered most essential are given the highest attention. The traits with the least significance are considered to have a lower priority compared to others. A characterization distribution was conducted. For instance, there are four features considered as high priority, and four traits considered as secondary priority. Moreover, two supplementary characteristics are classified as the third most important [26]. The researchers used the KDD'99 dataset for their experiment. The findings demonstrate that the hybrid

model attained a detection rate of 0.973, indicating a favourable outcome. The rate of false alarms was calculated to be 0.017 [27]. Researchers [28] extensively focus on database intrusion, namely the use of data mining techniques for detecting anomalies. In addition, the author employs advanced implementations of rules for associations using Trie trees. A feature selection model leveraging multilayer perception (MLP) was developed by the researchers for an IDS. The following model is integration of principal component analysis (PCA) and GA. To decrease the number of features to a primary feature space, the researchers used PCA [29]. The characteristics associated with the greatest eigenvalues were selected. The PCA-selected features may lack enough detection capabilities for the classifier. Hence, the researchers used GA to explore the primary feature space in order to pinpoint a subset that exhibits the utmost sensitivity. A thorough examination of the use of machine learning (ML) technologies in the intrusion detection system has been carried out by the authors [30]. In addition to providing a detailed analysis of several ML techniques for the Intrusion Detection System, the work addresses the use of machine learning in systems. This study demonstrates that one major issue in training machine learning models is insufficient or nonexistent traffic data. An artificial neural network (ANN)-based IDS model has been published by the authors [31]. Their method has a drawback in that the recommended model takes a long time to train. However, even in the absence of adding more agents to the present ones, the neural network's overall detection performance will remain constant. The two most popular feature selection methods, wrapper and filter, are covered by the researchers [32]. The effectiveness of two algorithms combined with a genetic algorithm-based selection strategy is compared in this research. The results show that when it comes to selecting characteristics, the Genetic Algorithm outperforms the Filter and Wrapper algorithms by a significant margin. Researchers [33] presented a paradigm for feature selection in IDSs. The aforementioned properties were selected using evolutionary algorithms, namely differential evolution (DE), particle swarm optimization (PSO) [25], and GA [26]. An assessment of the effectiveness of these algorithms was conducted. The validation method was carried out using the KDD Cup 99 dataset, together with a neural network and an SVM. The GA, PSO, and DE algorithms selected the following optimum attributes in the given order: 16, 15, and 13. They were selected among the forty-one characteristics that make up the dataset. The researchers concluded that the training process for DE takes around 1.62 seconds. According to their findings, the DE algorithm is considered the most precise method for categorization [34]. To be more specific, the categorization accuracy of DE is 99.75%. Using the Genetic Algorithm, researchers [35] provide an important method for selecting the required subset of features. They are convinced that feature selection may effectively eliminate superfluous components and have a significant impact on the subsequent development of effective categorization systems. A novel approach to feature selection is put forward by the authors [36]. It blends the mathematical intersection notion with the evolutionary algorithm. Conventional IDS frequently encounter challenges such as elevated rates of false positives, limited capability to adjust to novel attack patterns, and inefficiencies in handling substantial amounts of data. These issues require the creation of IDS that are more precise and effective. An essential factor in enhancing IDS performance is the efficient selection of features. It can enhance intrusion detection accuracy and decrease computing burden by recognizing and leveraging the most pertinent characteristics in network data. Evolutionary algorithms, drawing inspiration from the process of natural selection, demonstrate efficacy in improving intricate issues. They have the ability to systematically search for the most optimal features for intrusion detection, adjusting to new data and attack patterns in a step-by-step manner. The researchers used GA's [37] as a method to find potential intrusions in the network traffic. By looking at various connection properties, a set of classification criteria was created. Details such as the kind of network connection and how long the connection lasted. The

**Table 1. List of abbreviations.**

| GA | Genetic Algorithms |
|---|---|
| DT-SVM | Decision Tree-Support Vector Machines |
| FPR | False Positive Rate |
| TPR | True Positive Rate |
| FAR | False Alarm Rate |
| PSO | Particle Swarm Optimization |
| ACO | Ant Colony Optimization |
| IDS | Intrusion Detection System |
| MLP | Multilayer Perception |
| ML | Machine Learning |
| PCA | Principal Component Analysis |
| ANN | Artificial neural network |
| DE | Differential Evolution |
| kNN | k Nearest Neighbor |
| RF | Random Forest |
| DNN | Deep Neural Network |
| DE | Differential Evolution |
| EA | Evolutionary Algorithms |
| EC | Evolutionary Computing |

widely-known benchmark dataset KDD-Cup 99 was used to assess the suggested methodology. The authors in [38] presented a methodology that employs a genetic algorithm to ascertain the optimal deep neural network (DNN) architecture for binary classification in the context of network intrusion detection. This approach considers both the quantity of hidden layers and the quantity of neurons in each layer. The proposed results demonstrate that the DNN architecture exhibits superior performance compared to conventional machine learning methods, all the while requiring fewer computational resources. Current machine learning approaches face several issues, including probing and overfitting. Acquiring the necessary data to train these techniques requires a greater amount of time and effort. A significant number of contemporary machine learning algorithms lack the capability of autonomous online learning, thus requiring additional resources for training and processing. Following Table 1 is showing the tabular summary of related work. Following Table 2 Specific Strengths, Weaknesses, Limitations, and Challenges of Existing Solutions for Evolutionary Intrusion Detection System.

## 3 Proposed methodology

The following section consists of proposed methodology where GA has been used for feature selection using hybrid ML model, description of steps required for data set preprocessing followed by the algorithm.

### 3.1 Dataset preprocessing

The BoT-IoT [37] and UNSW NB 15 [38] datasets have been hybridized [39] for their appropriateness in tackling security issues. Individually, the BoT-IoT incursion dataset consists of 72 M data records and includes 45 characteristics. Furthermore, the intrusion dataset UNSW NB 15 has a total of 25 million records and includes 49 distinct characteristics. The composite dataset undergoes three pre-processing processes, including data cleaning, discretization, and normalization. The primary phase of data preparation. All variables are available in both

**Table 2. Related work.**

| Ref. No | Dataset | ML/DL Classifier | Parameters | Findings | Limitations |
|---------|---------|------------------|------------|----------|-------------|
| [45] | UNSW NB 15 and CICIDS | SVM, kNN, Xgboost | Accuracy, Precision and Recall | Effectiveness of genetic algorithms for reducing feature dimensions in IDS with high accuracy in feature reduction and improved detection rates | Computationally intensive and Longer training times |
| [46] | NSL-KDD | GD-ANN, DT | Accuracy, precision, recall and F1-Score | Enhanced optimization capability and Better stability in feature selection | Complexity in implementation and Risk of overfitting |
| [47] | CICIDS, NSL-KDD | DNN | Accuracy | Superior performance in large datasets and improved detection of novel attacks | Requires extensive computational resources and Complex parameter tuning |
| [48] | UNSW NB15 | DT, RF | Accuracy, Precision, FPR | Achieved best results for identification of attacks | - |
| [49] | BoT-IoT | RF, DT, XgBoost | Accuracy, Precision, Recall | Advanced preprocessing techniques | Fewer features used in the predictions of attacks |
| [50] | NSL-KDD, KDD-CUP99 | RF and DT | Accuracy, | Better feature extraction for high accuracy | Need resources and computing time to achieve results and to understand what each image means |

datasets. Impute the missing values for variables that are present in one dataset but not in the other. The hybrid dataset contains instances with missing values. Primary requirements include the process of data cleaning, which is the removal of missing and duplicate data. In order to get reliable and exact results, it is advisable to either leave any missing values empty or manually replace them with the mean attribute or the most likely value. Discretization, the second preprocessing procedure, is used to apply cut points to the merged dataset. The cut points partition the range of values for quantitative qualities into a limited number of intervals, guaranteeing that each interval retains strong class coherence. Various discretization approaches are used, such as entropy-based, chi-square, and wrapper-based methods [40]. In the following hybrid dataset, an entropy-based approach is used to quantify the uncertainty between features. Discretization can be computed as shown in (4):

$$T_r = \sum_{j=1}^{T} /Y_j/ \tag{4}$$

where r iterates over intervals, $T_r$ number of positive intervals within interval of F attribute and /Y/ is total number of labels. The normalized processing approach reliably and linearly maps the value range for each feature to intervals within [0, 1]. This streamlines arithmetic computation and obviates the need for numbers [41]. Z-Score normalization is the process of converting unstructured data into normalized values and can be computed as shown in (5):

$$u_i' = \frac{u_i - E^-}{std(E)} \tag{5}$$

where $u_i'$ normalized z-score values.

## 3.2 Proposed feature selection based on genetic algorithms-hybrid DT-SVM model

Various types of threats might potentially compromise a network, and an intrusion detection system can help in detecting and identifying them. Preprocessing operations are carried out after the training and testing of the data, during which the accuracy is estimated. The Hybrid DT-SVM Model-GAs, represents a sophisticated approach in machine learning that leverages

the complementary strengths of DT, SVM, and the optimization power of GAs. This model integrates the intuitive decision-making process of DTs with the robust classification capabilities of SVMs to handle both linear and complex non-linear patterns effectively. The integration begins with GAs selecting the optimal subset of features from the dataset. GAs operate through a process of evolution-inspired operations including selection, crossover, and mutation, continually refining the feature set by evaluating its performance using a fitness function typically based on the accuracy or the F1 score generated by the hybrid DT-SVM model. The DT component of the hybrid model excels in managing categorical data and explicit decision-making by recursively partitioning the data, thus making the model interpretable and efficient in handling varied data types. On the other hand, the SVM component is utilized for its capability to efficiently classify high-dimensional spaces and its effectiveness in scenarios where the separation margin between classes is vital. The synergy between DT and SVM in this hybrid model is crucial, as it allows for the leveraging of DT's handling of specific feature interactions while SVM provides robustness against overfitting, especially in complex classification landscapes. This hybrid model is trained on the selected features, ensuring that each algorithm plays to its strengths, thereby enhancing the overall predictive performance. The GA's role is critical as it dynamically searches for the most effective combinations of features that maximize model accuracy, thus directly influencing the model's capacity to generalize well on unseen data. The convergence of these techniques into a single framework presents a powerful tool for tackling complex predictive tasks in fields ranging from cybersecurity to medical diagnostics, where both precision and interpretability are paramount. The GA algorithm determines the selection of prospective subgroups to use, resulting in improved accuracy in forecasts. The algorithm identifies the most highly talented individuals in society to contribute to the advancement of future generations. The proposed methodology employs the GA [19] as the essential foundation for the feature selection method. The unique chromosomes make up the initial population. The sequence 110110111–00101101 indicates how characteristics are represented using binary encoding in the BoT-IoT and UNSW-NB15 datasets. The chromosomes are produced at random. However, the initial population size is limited to 100–150 in order to include a broad variety of attack types in both datasets. Prior studies have shown that an algorithm's computing time and complexity increase with increasing beginning population size. However, decreasing the initial population will have an adverse effect on the algorithm's performance and increase the likelihood that a poor solution will be chosen. Both of the original datasets are divided into distinct training and testing datasets using the K-Fold validation approach.

Use the same approach all through the teaching process. The crossover and mutation rates don't change during the experiment. The mutation will discover a path to investigate an unexplored region of the search domain (a novel configuration of weights) can be computed as shown in (6) [42]:

$$A_{i,j} = P_{i,j} + F * (P_{best,j} - P_{i,j} + P_{r_1,j} - P_{r_2,j}) \qquad (6)$$

where $A_{i,j}$ is mutant vector, $P_{best,j}$ is populations best individual, $P_{r1}j$ and $P_{r2}j$ are the individuals of randomly selected populations. F is the fitness function can be evaluated as shown in (7) [43]:

$$F = w_a * mutation + Crossovr - w_b * selection \qquad (7)$$

where $w_a$ and $w_b$ are the weights for SVM and DT. Once all the various mutant vectors have been computed, the next phase in DE involves generating an equal number of descendants as there are individuals in the population. The weight of each child $B_i$ will be derived exclusively in the population ($P_i$) or the mutated vector ($A_i$) from the current generation (j). The

subsequent content will be as shown in (8) [44]:

$$B_i, j[g][r, c] = A_{i,j}[g][r, c] \tag{8}$$

where g is the specific inside weight. Selection criteria is calculated as shown in (9) [45]:

$$A_{i,j+1} = A_{i,j} + P_{i,j} \tag{9}$$

Following the completion of the iteration, each chromosome is assessed by the Fitness Function according to the classification outcomes produced by the DT-SVM. When one or more of the following conditions is satisfied, the feature extraction process ends:

1) After doing the maximum number of predetermined iterations, the search is concluded.

2) Over ten consecutive generations, the maximal level of fitness remains constant.

## 3.3 Proposed methodology-feature selection based on GA and fitness function

In order to discover useful characteristics, the feature selection approach that is grounded on the principles of the GA has been approached. In the context of the GA, chromosomes has been used to represent unique combinations of features.

Fig 1 illustrates the proposed methodology. Algorithm 1 illustrates the working of proposed methodology.

Furthermore, the Fitness Function is used to evaluate each chromosome. Only the chromosome with the highest fitness value is eligible to proceed to the next stage of evolution. Within the whole collection of chromosomes, known as the original population size, the new chromosome will replace the existing one. When the evolutionary cycle ends, the GA outputs traits that are generally typical. Subsequently, the random forest algorithm is used to further pick features and classify the obtained findings. The DT-SVM algorithm is generally recognized as

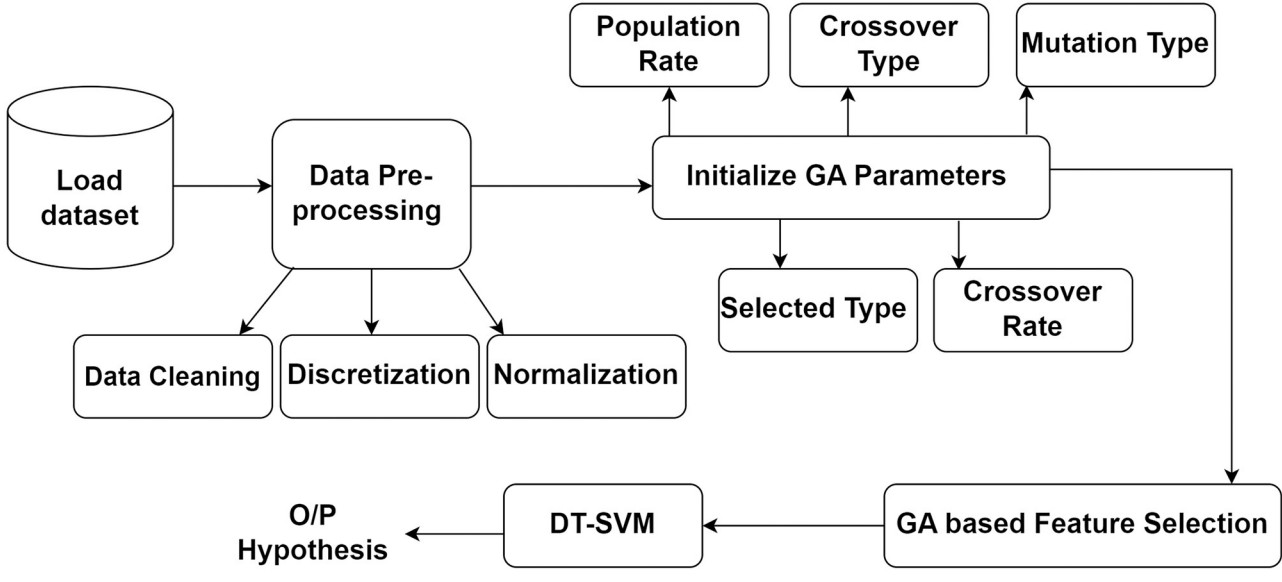

**Fig 1. Proposed methodology.**

a very efficient tool for managing complex data, regardless of whether it is used to binary or multi-class classification tasks.

In the genetic process that decides which chromosomes will survive, the Fitness Function is generally considered to be an important and crucial component. At the end of each evolutionary phase, the chromosome with the lowest score is swapped out with the one with the greatest score, as per the Fitness Function. Using a suitable fitness function to preserve chromosomes with high fitting values optimizes the genetic algorithm's iterative process. Earlier, researchers would choose subsets that had fewer features and better classification accuracy. The performance of the Intrusion Detection System would be negatively affected since some feature sets would produce more false alerts due to their failure to consider incorrect identification.

DTs iteratively divides the data space in order to categorise data. A decision tree consists of nodes that form a rooted tree, which is a directed tree with no incoming edges and a node designated as the "root." Every other node has exactly one incoming edge. A node, referred to as an internal or test node, has outward edges. The terminal, often referred to as decision nodes and leaves, represents the remaining nodes in a tree. Every internal node in a decision tree partitions the instances into two or more subsets depending on a unique outcome of the attribute values. The condition pertains to a numerical range for characteristics. An automated decision tree is produced from a given dataset. An optimum decision tree is obtained by minimising the error in generalisation. Constructing an ideal tree from the supplied data, as shown in [45], is challenging. Additional parameters, such as the number of nodes and average depths are taken as the minimum.

In 1999 Vapnik et al. [46] presented a Statistical Learning Theory and Structural Minimization, uses SVM to identify the optimal hyperplane that effectively separates positive data from negative ones for classification purposes. The hyperplane is defined as the one that maximises the margin among the training samples that are most comparable to it. The samples that are closest to the separating hyperplane are referred to as support vectors. After identifying the hyperplane, future samples may be categorised by determining the side of the hyperplane they belong to as shown in (10):

$$(y, f(x)) = max(0, 1y * f(x)) \tag{10}$$

where y is the true class label, and f(x) is the decision function can be computed as shown in (11).

$$f(x) = wx + b \tag{11}$$

w is the weight vector, x is the input feature vector, b is the bias term. The class prediction is based on the sign of f(x). A training data S = $(a_1, b_1 \ldots \ldots a_m, b_m)$ is given where $a_i$ is the input pattern at the ith instance and class label $b_i \in -1, 1$ for a two class problem with length k having k = $|y|$. If $y_i k = 1$ indicating $a_i$ assigned to class k, else not assigned to class. The optimization problem for k-th class is as shown in (12):

$$min_{w_k, c_k} = 1/2 ||w_k||^2 + C \sum_{i=1}^{N} Q_{ik} \tag{12}$$

concerning $z_i k \left( w_k^T a_i + b_k \right) \geq 1 - Q_{ik}, Q_{ik} \geq 0$ The parameters $w_k$ and $c_k$ define classifier at the k-th class as shown in (13):

$$f_k(a_i) = w_k^T a_i + b_k. \tag{13}$$

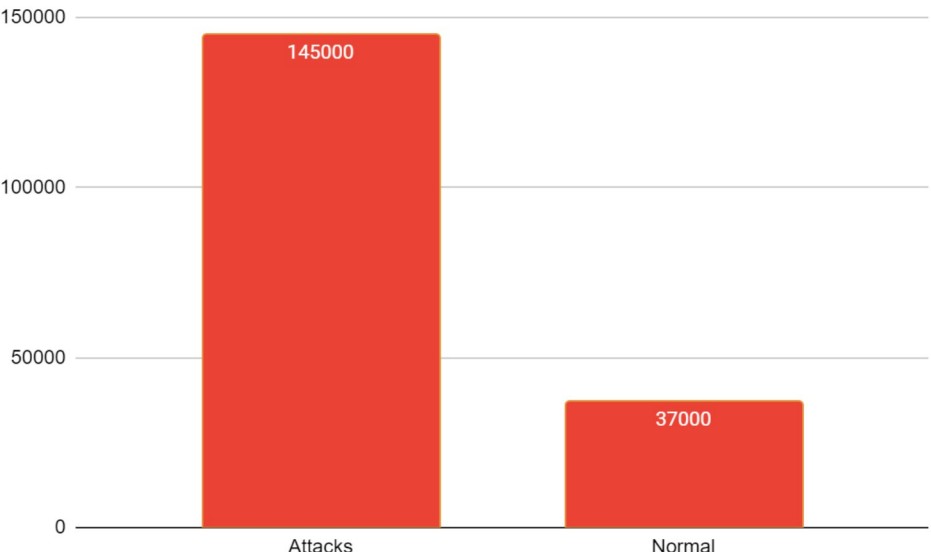

**Fig 2. No. of classes having normal and attack type.**

The classes having a set of binary classifiers are independent for label prediction $z$ for an instance which is unlabeled $a$ [15]. The component of k-th having vector label $z_k$ having value 1 if $f_k(a) > 0$ otherwise having value -1.

In the Fitness Function, $w_x$, $w_y$ represents the weights assigned to the accuracy of the DT and SVM represents the weights assigned to the False Positive Rate. The F1-score is a metric used to evaluate the correctness of a test. The harmonic mean is a mathematical measure that considers both accuracy and recall in order to calculate the performance of a classification model. The score of F1, which measures accuracy and memory, achieves its highest value at 1 (indicating flawless precision and recall) and its lowest value at 0. It is thought that the elevated False Positive Rate results in a False alert, potentially causing the Intrusion Detection System to classify regular network traffic as malicious. Our proposal aims to concurrently enhance the true positive rate (TPR) and reduce the false positive rate (FPR). Consequently, we include the FPR as a penalising factor in our Fitness Function, resulting in a decreased overall value of the Fitness Function when the FPR is large. At the conclusion of each iteration shown in Fig 2, the suggested Fitness Function assesses every chromosome, and only the ones with high scores are allowed to proceed to the next round of evolution.

**Algorithm 1** Proposed Algorithm: Feature selection based on GA

```
1: Hybrid dataset is loaded.
2: Apply Pre-processing techniques such as Data Cleaning, Discretiza-
   tion and Normalization using (4) and (5)
3: Load the dataset and split it into training and testing set.
4: Initialize GA Parameters.
i) Select two chromosomes with best fitness function as two parents.
d1 = randint(0, L) and d2 = randint(0, L)
Off_x = P_x and Off_y = Pj
Off_x[d1: d2] = P_y[d1: d2]
Off_y[d1: d2] = P_x[d1: d2]
ii) Calculate fitness function for each chromosome belongs to parent
population using fitness function
Fitness_(function) (Off_x)
Fitness_(function) (Off_y)
```

```
iii) Use the weights to train the DT-SVM over dataset
using (i + o) h_d + 2d.
iv) Compute the error
v) The fitness function is evaluated
5: Selected Features has been extracted.
6: Perform the k-cross fold stratified technique.
7: For the train attack set and feature set.
8: Apply DT-SVM
9: Obtain output
```

## 4 Results and discussion

The following section discusses the results and discussion for the experiment including the evaluation of time and space complexity.

### 4.1 Time and space complexity

There are five key components to the recommended approach. Every iteration of the genetic algorithm undergoes each of these procedures GA. For the initial population to be generated, the temporal complexity is $O(p_{size})$. For each of the eight selected features, a correlation matrix is computed within the fitness function that reflects the chromosomal length. The product of the length of the chromosome and the population size determines the time complexity of the fitness function. The temporal complexity of the roulette wheel, which is used to choose parents, is $O(p_{size})$. $O(p_{size})$ is the temporal complexity of crossover and mutation. When all variables are taken into account, the temporal complexity of the suggested technique may be expressed as $O(g((p_{size}) + p_{size} + c_{len}^2 + (p_{size}) + (p_{size})))$. Consequently, the overall temporal complexity of the suggested method is $O(g(p_{size} \times c_{len}^2))$. Let g represent the number of GA generations.

The population size and chromosomal length have a direct impact on the space complexity of the algorithm. The population and chromosome sizes are constant, with a population of 150 and a chromosomal size of 8. The average correlation for each chromosome is calculated by generating the correlation matrix. The space complexity of the correlation matrix is $O(c_{len}^2)$. Hence, the overall space complexity of the recommended approach may be calculated as the product of the population size and the length of the chromosome squared.

### 4.2 Analysis and discussion

The suggested methodology was implemented on each of the benchmark datasets, and a further set of experiments was done. The following experiment incorporates every step of the proposed process. These components include classification, a learning module based on genetic algorithms, and dataset preparation.

The suggested model is developed and is trained and verified using a 10-fold cross-validation technique. During each cycle, the dataset is randomly partitioned into ten distinct pieces. Four of these portions are used for model training, while the remaining portion is used for model validation. To streamline the training and validation process, the root mean square error produced during each iteration is saved separately. This enables the choice of an ideal subset of features from the data for classification purposes, rather than using every attribute of the data. The separate experiments were performed in this empirical investigation, with each experiment focusing on one of the two benchmark datasets. The learning module of proposed methodology was used to select a specific mix of characteristics for each trial. The presence of unique feature counts serves the purpose of establishing the ideal number of extracted features that effectively reflect the content of the dataset and thus enhance the learning process of

**Table 3. Specific strengths, weaknesses, limitations, and challenges of existing solutions for evolutionary intrusion detection system.**

| Ref. No | Approach | Strength | Weakness | Limitation | Challenges |
|---|---|---|---|---|---|
| [25] | Hybrid SVM-GA Model | High detection rate (0.973) and low false alarms (0.017) with reduced feature set. | May not perform as well with non-KDD'99 datasets. | Relies on optimal selection of features, which may not generalize across different attack vectors. | Ensuring robustness across diverse network environments and attack types. |
| [28] | MLP with PCA and GA | Reduces features to primary space, enhancing classifier efficiency. | PCA-selected features may lack detection capabilities without GA optimization. | Dependent on the initial PCA reduction quality. | Balancing dimensionality reduction and maintaining detection performance. |
| [31] | ANN-based IDS Model | Consistent detection performance without additional agents. | Long training times, which may impede timely deployment. | Performance capped without adding more agents; training data dependent. | Speeding up training processes while ensuring detection accuracy. |
| [48] | DE, PSO, GA Feature Selection (DT-RF) | High precision in classification with DE (99.75% accuracy); effective feature reduction. | Training time variability; DE takes 1.62 seconds, which may be slow for real-time applications. | Optimal attributes selection may vary significantly with dataset changes. | Achieving fast and accurate feature selection in dynamic real-time scenarios. |
| [38] | Genetic Algorithm for DNN Architecture | Superior performance with fewer computational resources | Might not adapt well to newer or unseen attack patterns. | Dependency on the genetic algorithm for optimal architecture can be resource-intensive. | Balancing computational efficiency with the adaptability of the DNN architecture. |
| [51] | MLP, J48 | Demonstrated potential in flow-based anomaly detection, providing good accuracy and efficiency in detecting anomalies within network traffic flows. | - | Limited by the reduction process and sampling, which may exclude some relevant anomalies or patterns. | The challenge of representing real-world traffic accurately, especially with synthetic or modified datasets. |

**Table 4. ML/DL classifiers used with IOT-IDS datasets using preprocessing techniques on features obtaining parameters.**

| Ref.No/ Year | Dataset | ML/DL Classifier | Pre-processing Techniques | Parameters |
|---|---|---|---|---|
| [38]/ 2021 | BoT-IoT | SVM | Feature extraction, Feature scaling | Accuracy-76.18% |
| [39]/ 2019 | BoT-IoT | NB, MLP | Feature selection and feature extraction | Accuracy-82.13% |
| [40]/ 2020 | UNSW-NB 15 | SVM | One hot encoding and feature selection | Accuracy-85.34% |
| [41]/ 2019 | BoT-IoT | CNN, MLP | - | Accuracy-87.34% |
| [42]/ 2019 | BoT-IoT | SVM | Data resolution, missing port numbers, data imbalance | Accuracy-90.02% |
| [43]/ 2022 | BoT-IoT | SVM | Missing values, entropy discretization and normalization | Accuracy-91.23% |
| [44]/ 2022 | BoT-IoT and UNSW NB-15 | SVM, MLP, DT | one-hot encoding and min-max normalization | Accuracy-94.23% |
| [51]/ 2024 | Winter and UNSW NB-15 | MLP DT | Standardization and normalization of features, handling missing values. | Precision, Recall, F1-Score, TPR, FPR |
| Our Work | BoT-IoT and UNSW NB-15 | GA based DT-SVM | K-Cross fold validation, min-max normalization, discretization, GA based feature selection and extraction, data cleaning, evolution parameters | Accuracy-96.12%-BoT-IoT and 92.06% for UNSW NB 15, Precision-90.23 FOR BoT-IoT and 92.36 for UNSW NB15, Recall-86.35 for BoT-IoT and 96.26 for UNSW NB15, F1-Score-88.25 for BoT-IoT and 94.27 for UNSW NB 15, FPR 2.47 for BoT-IoT and 1.45 for UNSW NB 15 along with Fitness Function |

**Table 5. Evolution parameters.**

| Parameter Name | Number |
|---|---|
| No. of features | 29, 47 |
| Chromosome length ($c_{len}^2$) | 8 |
| Initial Population ($p_{size}$) | 150 |
| Mutation Rate | 0.01 |
| Crossover Rate | 0.65 |
| Selection Type | Roulette Wheel |
| Crossover Type | Two-point |
| SVM kernel | Linear |
| cross fold validation (k) | 10 |

**Table 6. Fitness function parameters.**

| Parameter Name | Number |
|---|---|
| $w_x$ | 0.4 |
| $w_y$ | 0.6 |
| $w_z$ | 100 |

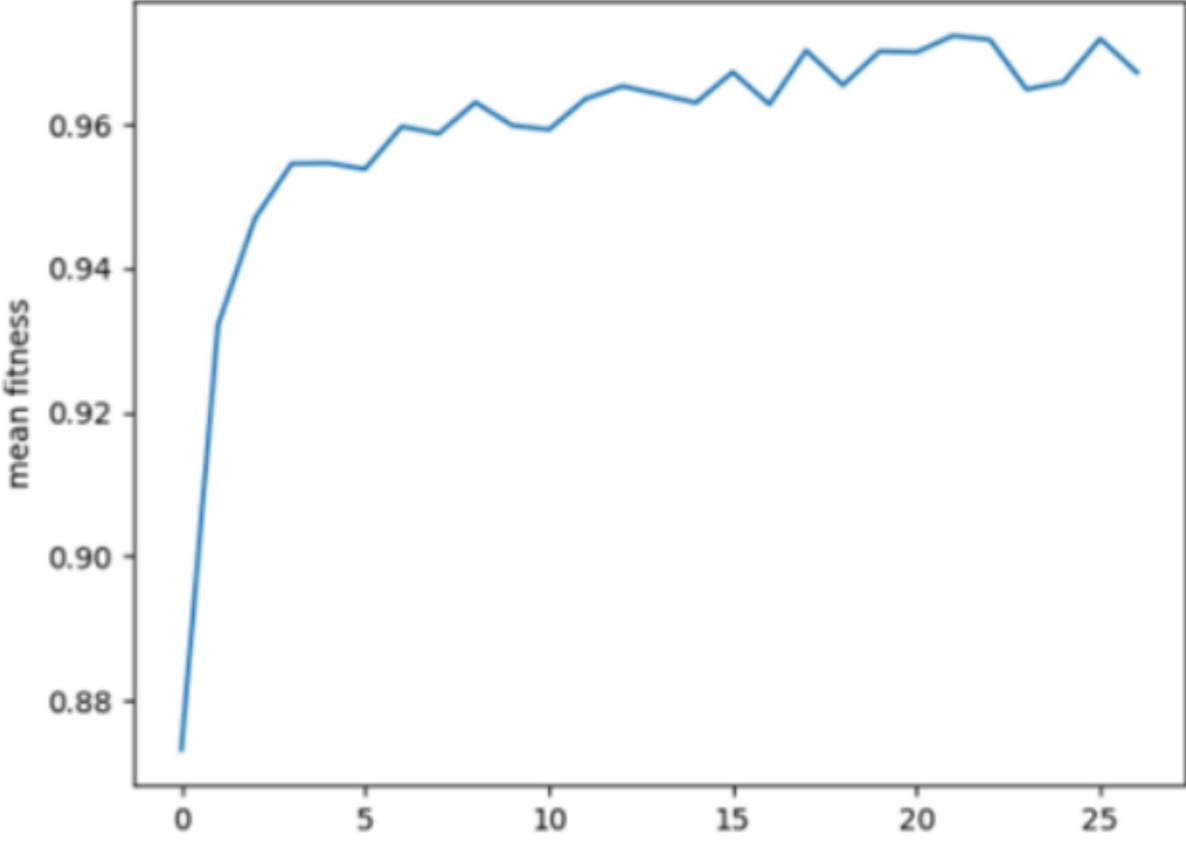

**Fig 3. Mean fitness in BoT-IoT.**

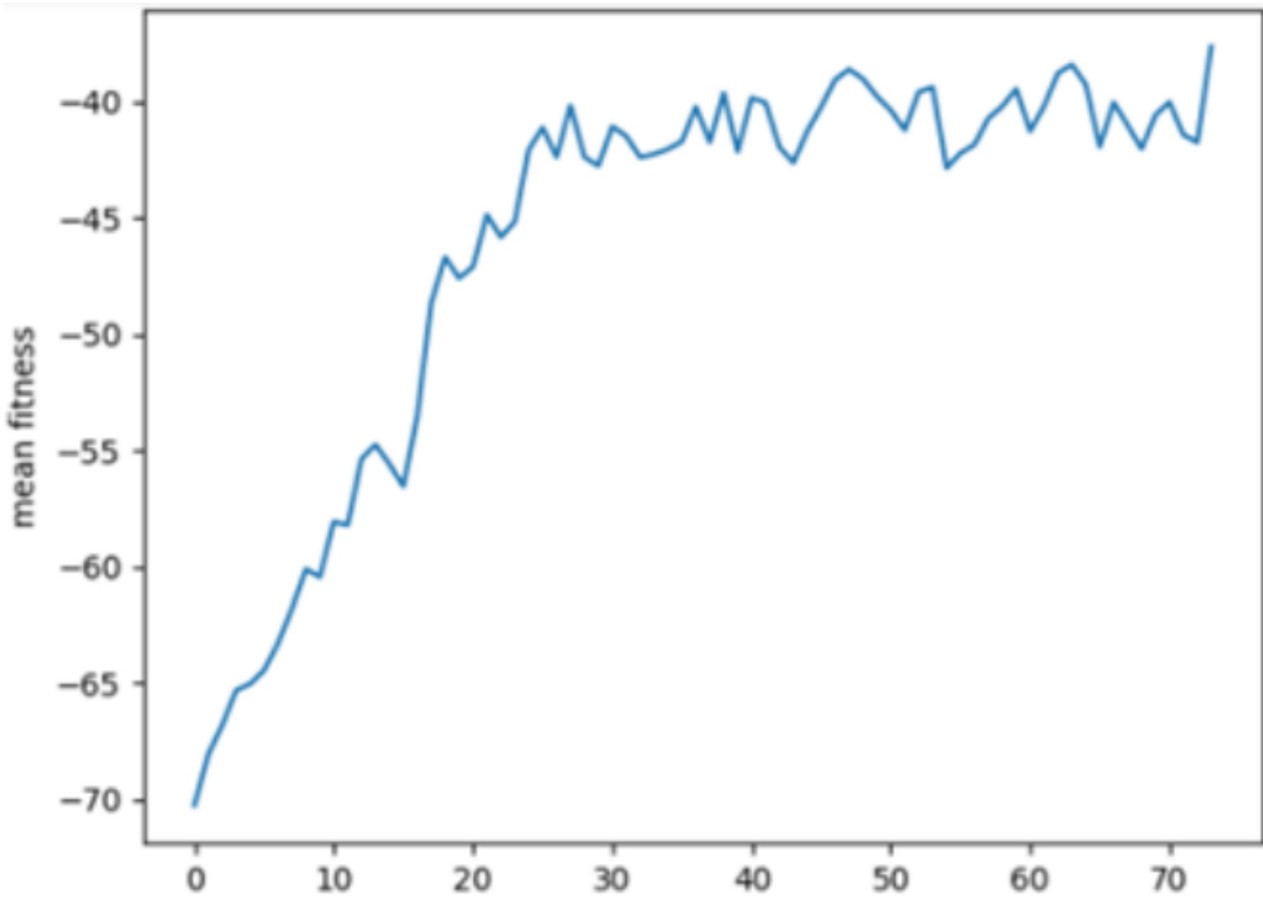

**Fig 4. Mean fitness in UNSW NB15.**

classifiers. Distinct experiments are performed, each involving the selection of eight, ten, and twelve features from the datasets, respectively. The Table 3 illustrates the comparative analysis of state of art with that of the proposed work.

## 4.3 Performance evaluation of hybrid DT-SVM model

The range of features employed in this study may span from the least to the greatest number of features selected it may involve an untested quantity of features. For this experiment, we have chosen the Bot-IoT dataset (which contains 29 features), and the UNSW NB-15 dataset (which contains 47 features), respectively. This section presents a performance study of attack detection but setting the values of evolution parameters and fitness function parameters using machine learning algorithms applied to both the datasets. Additionally, this section includes the evolution parameters as shown in Table 4 for the proposed DT-SVM-based attack type label categorization. K-fold stratified cross-validation is used to safeguard the model from overfitting, since it is a resampling approach utilized to evaluate the prediction capabilities of the model. K non-overlapping subsets of the same cardinality. The word "fold" is used to refer to the subsets that are produced [18]. In addition to k-1 subsets, the model is trained using a validation set to test its performance. The value of k is set to 10. In SVM a careful parameter tuning could improve the results achieved. DT maximize the correct classification of training data and to avoid overfitting pruning is applied to training tree and with large number of

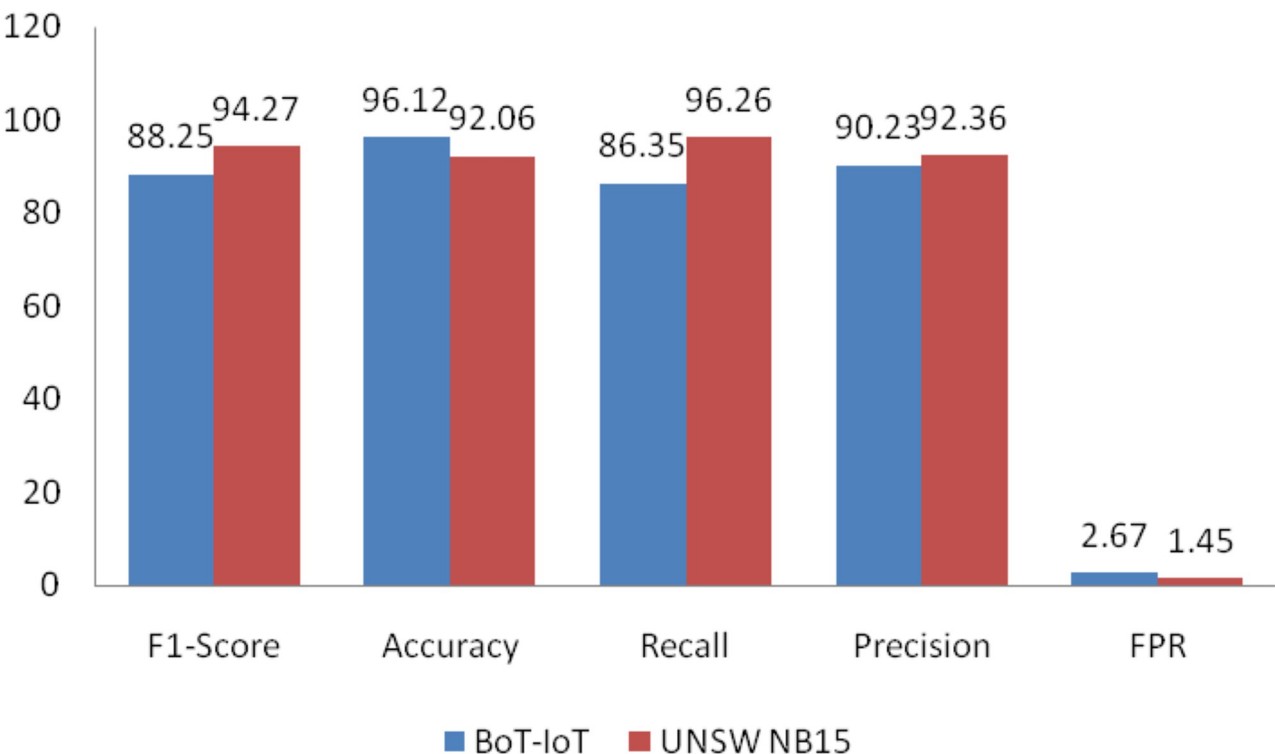

**Fig 5. Index evaluation in BoT-IoT and UNSW NB15.**

attributes it helped to achieve the best accuracy. There are 145000 instances for attack classes and 37000 instances for normal classes as shown in Fig 2.

## 4.4 Evaluation of fitness function

According to the Fitness Function, different parameter values as shown in Table 5 might potentially affect the fitness score. Mean fitness in this study is defined as the average performance score across multiple key metrics, including accuracy, precision, recall, and F1 score, calculated separately for each dataset. After conducting many trials, we concluded that the average fitness value had reached its highest point. when the value of $w_x$ was 0.4 and the value of $w_y$ was 0.6 as shown in Table 6. The framework was designed as a maximisation problem, with the value of $w_z$ being set to 100 to maximise the weight of FPR for best performance.

Figs 3 and 4 depicts the average fitness value for the BoT-IoT dataset and the UNSW-NB15 dataset.

The Y-axis represents the mean fitness values of each generation, while the X-axis represents the N×10th generation of the loop. This suggests that the population maintains accurately evaluated chromosomes, but the importance of chosen characteristics eventually stabilises.

Accuracy, Recall, Precision, FPR, and F1 → Score, FPR for both BoT-IoT and UNSW-NB15 dataset as shown in Fig 5.

The heatmap containing the features is shown in Fig 6. Fig 7 is shows the RMSE value of the proposed model.

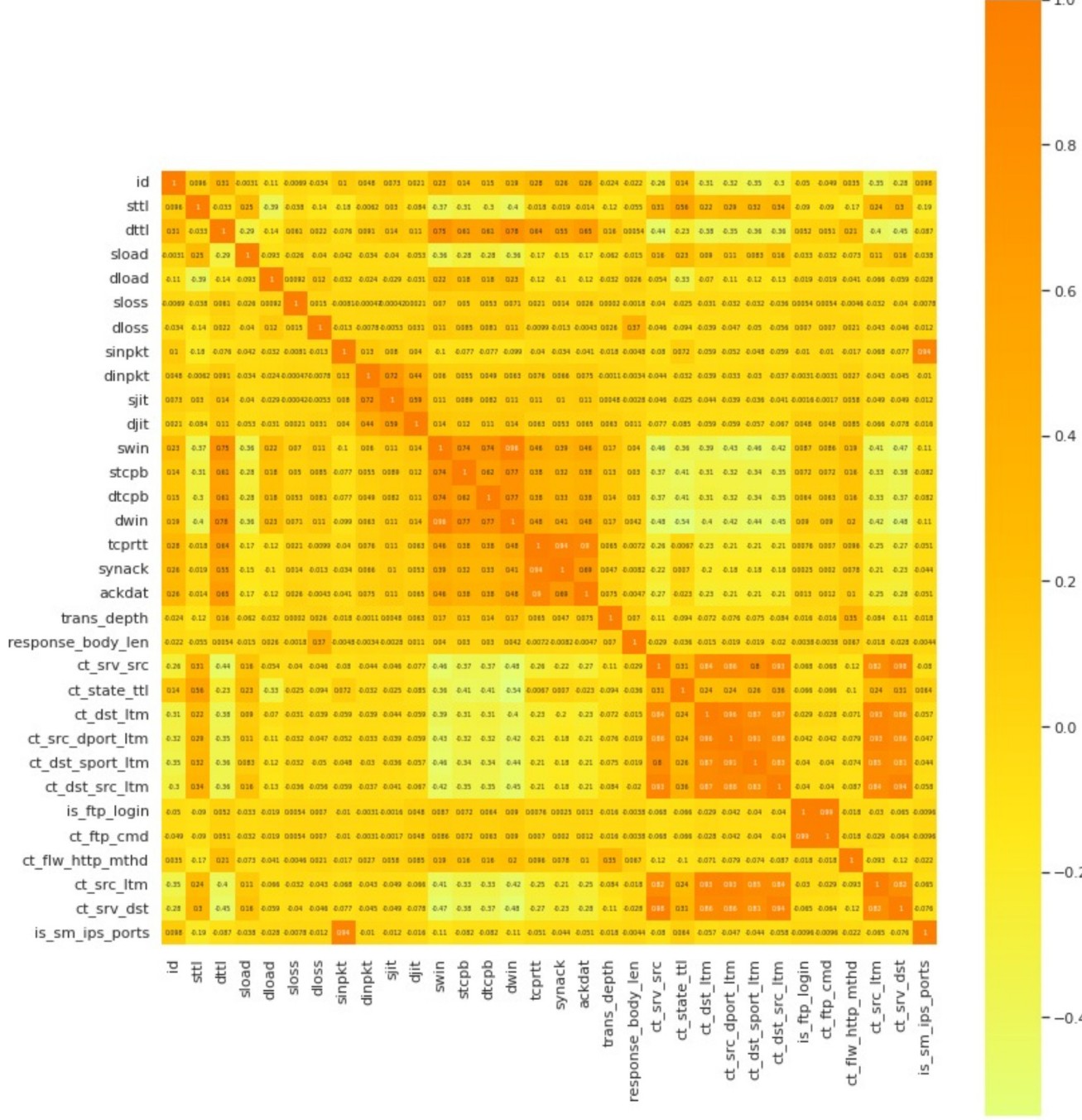

**Fig 6. Heatmap of the correlated features of the dataset.**

## 5 Conclusion

To sum up, a major breakthrough in cybersecurity has been made with the creation of an evolutionary machine learning algorithm for a reliable intrusion detection system. We have developed a system that can dynamically adapt to new cyberthreats by utilizing the power of evolutionary algorithms, strengthening the defences of networks against hostile assaults.

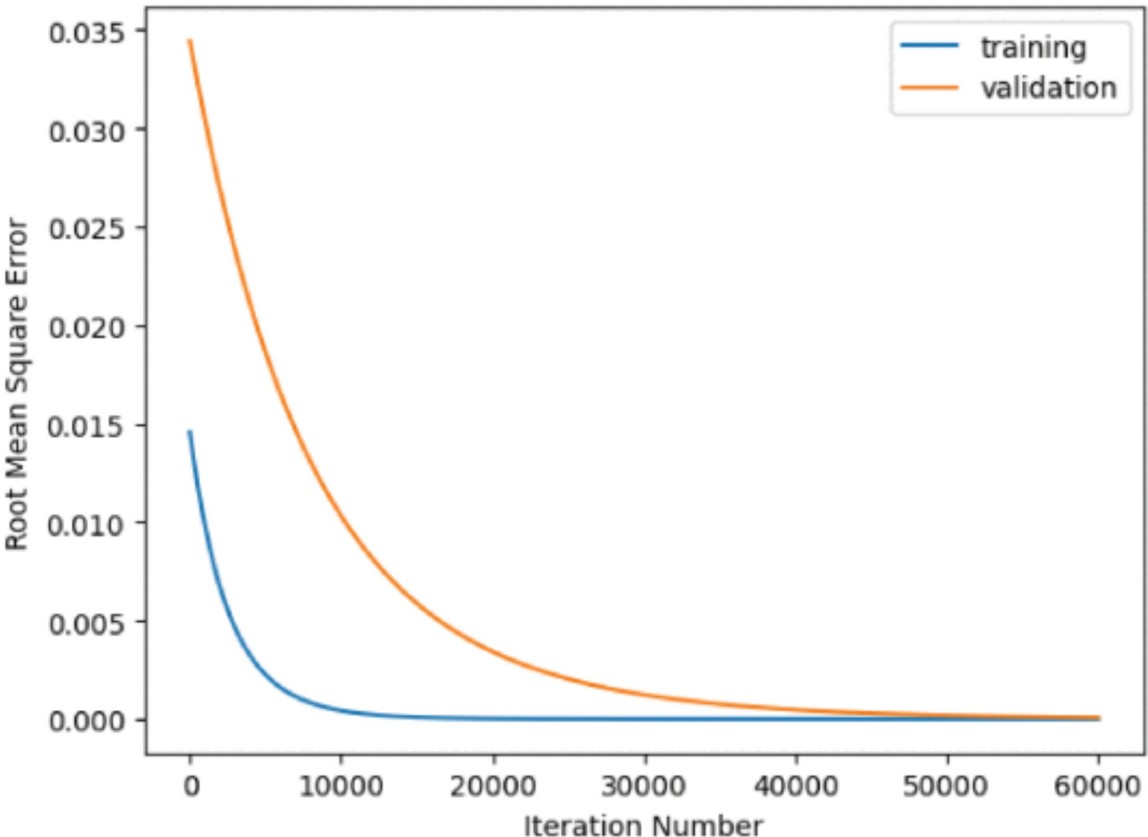

**Fig 7. Evaluation of RMSE of the proposed model.**

Proposed work in this manuscript exhibits the capacity to precisely recognize and categorize diverse forms of cyberattacks by incorporating machine learning methodologies, hence facilitating prompt and efficient remedial actions. The algorithm's evolutionary design guarantees ongoing optimization and enhancement, keeping up with changing threat environments. The UNSW NB15 and Bot-IoT intrusion datasets are used to assess mean fitness using machine learning classifiers. Additionally, the features are chosen using GA using the settings that are provided below. The item using a model that combines the Genetic algorithm with the DT and SVM is advised to get the best results. Furthermore, a newly developed Fitness Function incorporates the FPR as a penalty parameter in order to improve the identification of TPR and decrease the frequency of false alarms (FAR). The recall and the accuracy rates are given equal weight when using the F1-score. Our major objective is to reduce data dimensionality and optimise parameters to improve the accuracy and processing capability of DT and SVM. The item using a consistent number of selected attributes and making selections quickly are considered critical performance indicators. The use of BoT-IoT and UNSW NB 15 dataset has shown the effective results based on the proposed methodology in terms of accuracy, precision, recall and f1 score and FPR for both the datasets.

Numerous prospective opportunities exist for the ongoing research. The current methodology, which relies heavily on the GA, is inherently sluggish by virtue of its laborious replication processes and extensive iterations. Moving forward, the implementation of Evolutionary Strategy may be utilized to mitigate this issue and produce more rapid outcomes. In recent times,

binary chaotic genetic algorithms have demonstrated encouraging outcomes. In the future, these optimization strategies may be implemented for the identical assignment. In the phase of prediction, the proposed system implemented supervised learning by utilizing two hybrid classifiers. Moving forward, it is possible that computers will be capable of independently acquiring knowledge regarding novel attack types by employing unsupervised learning techniques, such as clustering.

## Acknowledgments

The authors would like to acknowledge the support of Prince Sultan University for paying the Article Processing Charges (APC) of this publication.

## Author Contributions

**Funding acquisition:** Maha Driss.

**Project administration:** Maha Driss.

**Validation:** Shalli Rani.

**Writing – original draft:** Ankita Sharma.

**Writing – review & editing:** Ankita Sharma.

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
