## [Decision Letter · Decision Letter 0]

25 Jun 2024

PONE-D-24-20436Hybrid Evolutionary Machine Learning Model for Advanced Intrusion Detection Architecture for Cyber Threat IdentificationPLOS ONE

Dear Dr. Rani,

Thank you for submitting your manuscript to PLOS ONE. After careful consideration, we feel that it has merit but does not fully meet PLOS ONE’s publication criteria as it currently stands. Therefore, we invite you to submit a revised version of the manuscript that addresses the points raised during the review process.

We look forward to receiving your revised manuscript.

Kind regards,

Raman Singh

Academic Editor

PLOS ONE

Additional Editor Comments:

As per the reviewer's report, the paper needs some improvements. It is advised to work on reviewer's comments. The reviewer has listed one reference to compare the results. If the authors think the work is relevant and comparison will add the readability and novelty of the paper, only then authors can decide on this aspect.

Reviewers' comments:

Reviewer's Responses to Questions

**Comments to the Author**

1. Is the manuscript technically sound, and do the data support the conclusions?

Reviewer #1: Partly

2. Has the statistical analysis been performed appropriately and rigorously? 

Reviewer #1: N/A

3. Have the authors made all data underlying the findings in their manuscript fully available?

Reviewer #1: No

4. Is the manuscript presented in an intelligible fashion and written in standard English?

Reviewer #1: Yes

5. Review Comments to the Author

**Reviewer #1: **The authors propose an approach to enhance the accuracy of IDSs through the use of hybrid evolutionary approach for feature selection. I find the idea interesting but do have the following comments:

- introduction and motivation are very vague and don't really provice specefic information on the conducted research. The motivation to apply this approach is not really clear. The introduction section is very general and doesn't provide enough details on the problem being investigated; what it is a problem? why it is a problem? why a new approach is needed. The main problem the authors try to address is the proper selection of features for machine learning algorithms used in IDS. So, i expect the authors to focus on that in their motivation and introduction.

- contribution is also not clear

- I also advice the authors to compare their approach to flow-based anomaly detection approaches like in L. Van Efferen and A. M. T. Ali-Eldin, "A multi-layer perceptron approach for flow-based anomaly detection," 2017 International Symposium on Networks, Computers and Communications (ISNCC), Marrakech, Morocco, 2017, pp. 1-6, doi: 10.1109/ISNCC.2017.8072036.

- more results on the attacks classification accuracy are needed

-some other points:

*the use of abbreviations before introducing them. I suggest adding a symbols table

*pg 13 line 6: ''The following prototype is integration of principal component analysis (PCA) and GA.'' I cant find the prototype the authors refer to

6. PLOS authors have the option to publish the peer review history of their article (what does this mean?). If published, this will include your full peer review and any attached files.

Reviewer #1: No

---

## [Author Response · Author response to Decision Letter 0]

2 Jul 2024

TITLE: An Evolutionary Machine Learning Algorithm for Robust Intrusion Detection System for Identification of Cyber Attacks

Authors : Ankita Sharma, Shalli Rani, Maha Driss

ankita.sharma@chitkara.edu.in, shalli.rani@chitkara .edu.in, mdriss@psu.edu.sa

Dear Editors and Reviewers: 

We are thankful to you for spending your valuable time for making a review and for constructing the comments on our manuscript. These comments are valuable and very helpful for revising and improving our paper. We have studied comments carefully and have made correction as marked in the revised manuscript. We have tried our best to address the mentioned comments to revise our manuscript in the hope that these revisions will meet your requirement. The following changes have been made in the manuscript as per the received comments. 

*The changes made in the manuscript as per received comments have been highlighted in red color.

Reviewer #1: The authors propose an approach to enhance the accuracy of IDSs through the use of hybrid evolutionary approach for feature selection. I find the idea interesting but do have the following comments:

Comment 1:- introduction and motivation are very vague and don't really provice specefic information on the conducted research. The motivation to apply this approach is not really clear. The introduction section is very general and doesn't provide enough details on the problem being investigated; what it is a problem? why it is a problem? why a new approach is needed. The main problem the authors try to address is the proper selection of features for machine learning algorithms used in IDS. So, i expect the authors to focus on that in their motivation and introduction.

Response 1: As per the comment whole introduction and motivation has been updated in the paper. Also, added “The following paper presents a Hybrid Evolutionary Machine Learning Model that is being designed to improve the capabilities of advanced IDS architectures. By leveraging the strengths of EA and ML techniques, this hybrid model aims to provide a more resilient and adaptive solution for cyber threat identification. EA offer robust search capabilities and adaptability, while ML techniques excel in pattern recognition and predictive analytics. The combination of these methodologies results in a powerful tool that can evolve and improve over time, adapting to new and emerging threats in the cybersecurity landscape.” In the last of intro section.

Motivation

The motivation behind this research stems from the increasing complexity of cyber threats and the limitations of existing intrusion detection systems. Traditional IDS often rely on static rule-based approaches, which are insufficient in detecting novel and sophisticated attacks. Furthermore, machine learning-based IDS, while more dynamic, can suffer from issues such as high false positive rates and the inability to adapt to rapidly changing threat environments.

Also, conventional models frequently fail to detect intricate or groundbreaking attacks, resulting in high rates of false negatives and the introduction of vulnerabilities in the system. To overcome the limitations of traditional system the proposed approach focuses on enhancement in the detection and management of a wide range of cyber threats, hence reducing the likelihood of security breaches, by highlighting the algorithm's ability to adapt to the threat landscape which motivated us to integrate ML with EC.

By combining EA with machine learning, the proposed hybrid model seeks to address these challenges by providing a more adaptive and accurate intrusion detection system. EA can explore a vast search space and optimize detection strategies over time, while machine learning models can continuously learn and update based on new data. This hybrid approach aims to reduce false positives, enhance detection accuracy, and provide a scalable solution

Comment 3:- contribution is also not clear

Response 3: As per the comment it has been updated “This study employs the Bot-IoT and UNSW-NB15 intrusion datasets to evaluate mean fitness using ML classifiers. An innovative fitness function is constructed for the GA to rank features effectively. The feature selection process is guided by GA, with specific parameters tailored for optimal performance. 

To achieve the best possible outcomes, the model integrates the GA with DT and SVM. Additionally, the FPR is incorporated as a parameter in the newly created fitness function to simultaneously reduce FAR and enhance the detection of TPR. A high FPR can lead to many benign activities being incorrectly classified as threats, resulting in unnecessary alerts and operational inefficiencies. This disrupts user trust and increases the operational burden due to the need to investigate each false alarm, thus acting as a penalty on the system’s usability and resource management.

The study develops a genetic algorithm-based feature selection methodology that leverages evolutionary computing for feature selection in intrusion detection systems. The F1-score is used to balance the importance given to both the recall rate and the accuracy rate. The F1 score harmonizes recall and accuracy, addressing potential imbalances that might lead to misleading interpretations of a model's accuracy. The main goal is to enhance the precision and computational capacity of DT and SVM by optimizing parameters and reducing data dimensionality.”

Comment 4:- I also advice the authors to compare their approach to flow-based anomaly detection approaches like in L. Van Efferen and A. M. T. Ali-Eldin, "A multi-layer perceptron approach for flow-based anomaly detection," 2017 International Symposium on Networks, Computers and Communications (ISNCC), Marrakech, Morocco, 2017, pp. 1-6, doi: 10.1109/ISNCC.2017.8072036.

Response 4: As per the comment added the same as Reference 51 also added the entry of the same in Table 2 and 4 for comparison.

Comment 5: more results on the attacks classification accuracy are needed

Response 5: As per the experiment done I have evaluated the heatmap, graphs and tables which I have already put in the result section. 

Comment 6: -some other points:

*the use of abbreviations before introducing them. I suggest adding a symbols table

Response 6: The same Table has been added as Table 1.

Comment 7: *pg 13 line 6: ''The following prototype is integration of principal component analysis (PCA) and GA.'' I cant find the prototype the authors refer to

Response 7: Sorry for the misinterpretation, The word prototype mentioned in Related Work meaning the authors discussed about the “model”, I wrote prototype for the other word for model. Therefore I have made it correct and updated the word “model” only.

---

## [Decision Letter · Decision Letter 1]

19 Jul 2024

Hybrid Evolutionary Machine Learning Model for Advanced Intrusion Detection Architecture for Cyber Threat Identification

PONE-D-24-20436R1

Dear Dr. Rani,

We’re pleased to inform you that your manuscript has been judged scientifically suitable for publication and will be formally accepted for publication once it meets all outstanding technical requirements.

Kind regards,

Raman Singh

Academic Editor

PLOS ONE

Additional Editor Comments (optional):

Reviewers' comments:

Reviewer's Responses to Questions

**Comments to the Author**

1. If the authors have adequately addressed your comments raised in a previous round of review and you feel that this manuscript is now acceptable for publication, you may indicate that here to bypass the “Comments to the Author” section, enter your conflict of interest statement in the “Confidential to Editor” section, and submit your "Accept" recommendation.

Reviewer #1: All comments have been addressed

2. Is the manuscript technically sound, and do the data support the conclusions?

Reviewer #1: Yes

3. Has the statistical analysis been performed appropriately and rigorously? 

Reviewer #1: N/A

4. Have the authors made all data underlying the findings in their manuscript fully available?

Reviewer #1: Yes

5. Is the manuscript presented in an intelligible fashion and written in standard English?

Reviewer #1: Yes

6. Review Comments to the Author

Reviewer #1: I would like to thank the authors for addressing my comments and further I have no other comments.

7. PLOS authors have the option to publish the peer review history of their article (what does this mean?). If published, this will include your full peer review and any attached files.

Reviewer #1: No

---

## [Editor Report · Acceptance letter]

26 Jul 2024

PONE-D-24-20436R1 

PLOS ONE

Dear Dr. Rani, 

I'm pleased to inform you that your manuscript has been deemed suitable for publication in PLOS ONE. Congratulations! Your manuscript is now being handed over to our production team.

Kind regards, 

on behalf of

Dr. Raman Singh 

Academic Editor

PLOS ONE